# Angio Cone-Beam CT (Angio-CBCT) and 3D Road-Mapping for the Detection of Spinal Cord Vascularization in Patients Requiring Treatment for a Thoracic Aortic Lesion: A Feasibility Study

**DOI:** 10.3390/jpm12111890

**Published:** 2022-11-11

**Authors:** Pierre-Antoine Barral, Mariangela De Masi, Axel Bartoli, Paul Beunon, Arnaud Gallon, Farouk Tradi, Jean-François Hak, Marine Gaudry, Alexis Jacquier

**Affiliations:** 1Department of Radiology, CHU Timone, AP-HM, 264, Rue Saint-Pierre, 13005 Marseille, France; 2Department of Vascular Surgery, CHU Timone, 264, Rue Saint-Pierre, AP-HM, 13005 Marseille, France; 3Aortic Center, CHU Timone, AP-HM, 264, Rue Saint-Pierre, 13005 Marseille, France; 4CRMBM-UMR CNRS 7339, Aix-Marseille University, 27, Boulevard Jean Moulin, CEDEX 05, 13385 Marseille, France; 5Department of Visceral and Vascular Radiology, Centre Hospitalier Universitaire de Clermont-Ferrand, Clermont-Ferrand, France Aortic Center, CHU Timone, AP-HM, 264, Rue Saint-Pierre, CEDEX 1, 13005 Marseille, France; 6Department of Neuroradiology, CHU Timone, AP-HM, 264, Rue Saint-Pierre, 13005 Marseille, France

**Keywords:** interventional radiology, preconditioning, endovascular

## Abstract

Background: Spinal cord ischemia is a major complication of treatment for descending thoracic aorta (DTA) disease. Our objectives were (1) to describe the value of angiographic cone-beam CT (angio-CBCT) and 3D road-mapping to visualize the Adamkiewicz artery (AA) and its feeding artery and (2) to evaluate the impact of AA localization on the patient surgical strategy. Methods: Between 2018 and 2020, all patients referred to our institution for a surgical DTA disorder underwent a dedicated AA evaluation by angio-CBCT. If the AA feeding artery was not depicted on angio-CBCT, selective artery catheterization was performed, guided by 3D road-mapping. Intervention modifications, based on AA location and one month of neurologic follow-up after surgery, were recorded. Results: Twenty-one patients were enrolled. AA was assessable in 100% of patients and in 15 (71%) with angio-CBCT alone. Among them, 10 patients needed 3D road-mapping-guided DSA angiography to visualize the AA feeding artery. The amount of contrast media, irradiation dose, and intervention length were not significantly different whether the AA was assessable or not by angio-CBCT. AA feeding artery localization led to surgical sketch modification for 11 patients. Conclusions: Angio-CBCT is an efficient method for AA localization in the surgical planning of DTA disorders.

## 1. Introduction

Neurological complications, such as spinal cord injury, remain a major concern in the treatment of descending thoracic aortic (DTA) disease. Spinal cord ischemia arises in between 1% and 8% of patients after treatment with DTA [1,2,3]. Several methods to decrease the spinal cord injury rate have been described, including cerebrospinal fluid (CSF) drainage, motor somatosensory-evoked potential, or surgical reimplantation of the feeding artery of the Adamkiewicz artery (AA) [4,5].

The need for assessing the anatomical location of the feeding artery of the AA before DTA treatment is debated in the literature for several reasons. The variability of its anatomical origin is high, most commonly found between T8 and L1, and originates from the left intercostal or lumbar artery in 70% to 85% of cases [6,7]. Additionally, the feeding artery of the AA is a small artery, and its anatomical location might be hidden with the deformation of the aneurysmal aorta and aortic thrombus. However, a lower rate of postoperative neurologic complications is observed in patients with DTA surgical repair if they benefit from previous AA detection and preservation [8]. If the AA feeding artery is covered in TEVAR, CSF drainage is associated with a lower incidence of symptomatic spinal cord ischemia [9]. There is no consensus regarding spinal cord ischemia prevention before surgical treatment of the descending thoracic aorta. The anatomical location of the AA might help in decision-making before treatment. Selective DSA of each patent intercostal artery could be very challenging in the case of large aortic aneurysms. New angiographic techniques such as angio-CBCT might help in detecting small arteries such as the AA in large aneurismal vessels.

Several invasive and noninvasive techniques to assess the AA location have been described [10,11,12,13,14]. Noninvasive assessment methods involving the use of magnetic resonance (MR) angiography and multidetector row CT angiography have recently been employed [15,16,17]. However, their wide adoption can be limited because of patient morphology. On the other hand, digital subtraction angiography (DSA) is challenging to perform, as it is difficult to catheterize the intercostal artery ostium in an aneurysmal sac.

The periprocedural use of cone beam CT (CBCT) has been described in many interventional radiology procedures [18]. The arm of the angio-suite can perform localized CT acquisition by rotating around the patient in a cone-shaped beam. The acquisition, coupled with vessel angiography, is called angio-CBCT. It allows small vessel detection with a higher spatial resolution than CT–angiography and MRI acquisition. Additionally, angio-CBCT acquisition can be used as a 3D road-mapping mask for the rest of the procedure. It was found to be efficient in the detection of injured vessels for emergency embolization, transarterial chemoembolization guidance, or the evaluation of spinal arteriovenous fistulas [19,20,21]. However, there is no evaluation of this technique in the preprocedural visualization of the AA feeding arteries before DTA disease treatment. Therefore, our goals were to (1) describe the value of contrast angio-CBCT and 3D road-mapping in locating the feeding artery of the AA in patients with DTA disease requiring surgical treatment; (2) quantify the total amount of contrast media, irradiation dose and intervention length required to locate the feeding artery of the AA; (3) assess the impact of the preprocedural location of the feeding artery of the AA on the treatment strategy; and (4) report the rate of one month postprocedural neurologic complications.

## 2. Materials and Methods

### 2.1. Study Design

This single-center nonrandomized retrospective study included all patients being followed in our center for DTA disorders with planned thoracic or thoracoabdominal aortic repair with a risk of spinal cord infarction from February 2018 to April 2020 in the Centre Aorte Timone. The exclusion criteria were patients referred for emergency and contraindications for iodine injection. All patients included during this period underwent angiography with angio-CBCT acquisition after iodine injection and, if needed, selective opacification of selected patent intercostal artery-guided 3D road-mapping to detect AA and AA feeding arteries. In the second phase, we evaluated our attitude toward surgical treatment modification after AA detection. Ethical review and approval were not applicable for this retrospective study.

### 2.2. Diagnostic Angiography Procedure

All angiographic procedures were performed in an angiography room (Discovery IGS 730, General Electric, Buc, France) under local anesthesia. The first phase of the intervention was always angio-CBCT aortic acquisition. We performed a femoral approach with the Seldinger technique (5F introducer sheath, Terumo Cardiovascular Systems Corporation, MI, USA). A multipurpose 5F pig tail catheter was placed at the level of T9. Angio-CBCT acquisition was then performed during a breath hold at a rotation speed of 40°/s and during contrast media injection. A diluted bolus was used with 50% saline and 50% iodixanol 320 (Visipaque, Gerbet, Aulnay-sous-Bois, France). Seventy cc of this mixture was injected with the following parameters: injection speed = 10 cc/s, delay between injection and angio-CBCT acquisition: between 3 to 4 s, according to the volume of the aneurysm and left to the judgment of the radiologist (Figure 1). In the case of a lack of opacification of the feeding artery of the AA, a second angio-CBCT was performed at a different level of the aorta depending on the aortic lesion anatomy (mostly involving a thoracic or abdominal disorder). Following the acquisition, images were processed on a dedicated work station AW Volume Share 4.6 (GE Healthcare, Chicago, IL, USA) to detect the feeding artery of the AA using a double oblique view within MPR reconstruction and Volume Rendering (Figure 1).

The AA was considered assessable on angio-CBCT and angio-CBCT positive if we visualized an AA artery defined as “a collateral artery of the radiculomedullary artery running obliquely along the anterior surface of the spinal cord with a classic hairpin turn connection to the anterior spinal artery” (Figure 2).

If the feeding artery of the AA was clearly defined, exploration was considered complete.

The definition of the feeding artery of the AA was a continuous vascular route for the anterior spinal artery, the AA, the radiculomedullary artery, the posterior branch of the intercostal (or lumbar) artery, the intercostal (or lumbar) artery and the aorta.

If the feeding artery was not clearly visualized on angio-CBCT alone, selective DSA angiography catheterization of the arteries at the probable origin of the AA was performed. In this case, we used 3D road-mapping extracted from angio-CBCT, reconstructed in volume rendering reconstructions, with the localization of the intercostals and lumbar artery ostia to facilitate catheterization (Figure 3).

AA was considered not assessable on angio-CBCT and angio-CBCT negative if no artery meeting the AA definition was visible, even after 2 angio-CBCT procedures. In this situation, selective DSA angiography catheterization of the arteries was performed. The 3D road-map extracted from the angio-CBCT acquisition was also used, but in contrast to the previous situation, catheterization was not guided by AA detection, and all intercostal and lumbar arteries from the region were catheterized. Selective angiography of patent intercostal/lumbar arteries requires different types of catheters, and the choice of catheter was left at the discretion of the radiologist. A manual injection of 4 cc of contrast media iodixanol 320 (Visipaque, Gerbet, Aulnay-sous-Bois, France) was used for selective angiography. Selective DSA was considered successful when the feeding artery of the AA was located on the DSA. A procedure was considered unsuccessful if angio-CBCT and DSA failed to locate the feeding artery of the AA.

The location of the ostium of the intercostal or lumbar artery, which feeds the AA, was highlighted on the volume rendering reconstruction of angio-CBCT or on the preprocedural CT acquisition. These images were used during multidisciplinary discussions to define the surgical procedure.

### 2.3. Surgical Management and Neurological Follow-Up

Treatment decisions were standardized and validated in a multidisciplinary team including vascular surgeons, cardiologists, anesthesiologists, and radiologists. Concerning endovascular repair, if the AA arose near the stent graft ends, the preservation of the ostium was maintained by stent graft length reduction. If the AA arose in the aneurismal area and needed to be covered by TEVAR, preventive CSF drainage was performed. In the case of open surgical repair, surgical reimplantation of the AA was performed. All these procedure modifications based on AA localization were discussed and recorded during multidisciplinary team meetings. One month after aortic intervention, a follow-up examination recorded early neurological complications. All types of neurologic symptoms were recorded: spinal cord ischemia, paresthesia or motor disorders of the lower limb, and Parkinson’s syndrome. The amount of iodine contrast injected, irradiation dose, fluoroscopy time, and procedure length were gathered. The continuous and categorical variables are described by the mean, standard deviation (SD) and range or median (Q1-median-Q3) and range, and n (%). The Mann Whitney U test was performed to evaluate both groups.

## 3. Results

A total of 21 patients were included in the study. Patient demographic and clinical characteristics are summarized in Table 1. All patients were able to benefit from the described diagnostic procedure with angio-CBCT. The average duration of the procedure was 48.7 ± 19.7 min for all 21 cases. In two cases, medullary arteriography was coupled to another intervention in the same operating time. The first was iliac stenting for aneurysmal exclusion, and the second was subclavian artery occlusion. Both interventions were performed in patients for whom angio-CBCT was positive and no extra DSA selective catheterization was necessary. The doses of contrast media and irradiation were combined in these two cases. No complications were noted during the procedure.

AA was assessable on angio-CBCT in 15/21 (71%) cases. Among them, the feeding artery was undoubtedly visible in 5/15 patients (33%). For the remaining 10/15 patients (67%), DSA selective angiography was needed to confirm the feeding artery. The AA was not assessable on angio-CBCT in 6/21 patients (29%). Three-dimensional road-mapping-guided DSA angiography of all intercostals and lumbar arteries finally helped to visualize the AA and the feeding artery in 6/6 patients (100%). These six patients with negative angio-CBCT had bulky aneurysms with high diameters (64.6 ± 9.3 versus 60.33 ± 7.6; *p* = 0.28) causing major flow turbulence after iodine injection. Figure 4 summarizes the study flow-chart.

The median and interquartile irradiation dose was 25.5 (20.0–40.0) Gy/cm^2^ for the positive angio-CBCT group and 70.4 (30.8–96.9) Gy/cm^2^ for the negative angio-CBCT group (*p* = 0.16) (Table 2). There was no significant difference in fluoroscopy time or total procedure time depending on the angio-CBCT result (Table 2). The difference in iodine contrast dose was not statistically significant. In two cases, medullary arteriography was coupled to another intervention in the same operating time. The first was iliac stenting for aneurysmal exclusion, and the second was subclavian artery occlusion. Both interventions were performed in patients for whom angio-CBCT was positive and no extra DSA selective catheterization was necessary. The doses of contrast media and irradiation were combined in these two cases.

The anterior spinal artery originated in 70% of cases from a left intercostal or lumbar artery and in 25% of cases from the 11th left intercostal artery. These data are presented in Table 3 [7].

After presurgical planification, 16 out of 21 patients underwent aortic surgical repair (Table 4). Eleven patients were treated with endovascular surgical repair (TEVAR), and five were treated with open surgery. The identification of the AA artery led to eight modifications of the surgical strategy for the endovascular population: decision for stent graft length reduction in five (45.4%) of the patients to preserve the AA feeding artery ostium and/or CSF monitoring for seven patients (64%). In the open surgery group, bridging was chosen for three patients (60%) and/or CSF monitoring/drainage in two (40%) patients after the visualization of the AA feeding artery.

One month of follow-up revealed a unique severe neurologic complication described as a spinal cord injury with permanent paraplegia. The patient was selected for open surgery, and the feeding intercostal artery of the AA was reimplanted in association with CSF drainage. During surgery, the patient suffered circulatory arrest due to occlusion of the interventricular artery. The patient was resuscitated but exhibited permanent paraplegia in the recovery room.

## 4. Discussion

Angio-CBCT and selective DSA guided by 3D road-mapping allowed the location of the feeding artery of the AA in 100% of the patients included in the present study. Angio-CBCT decreases the need for selective catheterization. Kieffer et al. described neurological complications after selective catheterization of the spinal or intercostal artery [10,22]. These types of complications were not reported in our series, and their exact prevalence is difficult to evaluate with the use of up-to-date selective catheters. Three-dimensional road-mapping allowed the location of the patent ostium of the intercostal artery for selective catheterization. In our series, the addition of selective catheterization of the intercostal or lumbar artery did not significantly increase the procedure length, showing that the use of image 3D road-mapping and location of the patent artery might simplify image acquisition and AA location. The presence of an aneurysmal sack thrombus may promote ostial occlusion of the intercostal arteries arising from the aneurysm [23,24]. The precise surgical location of the artery that feeds the AA plays a role in planning the surgical strategy in our department. Matsuda reported that the estimated incidence of permanent and transient spinal cord injury was 3.7% in all TEVAR patients, 6.0% when part or all of the distal aorta was covered and 12.5% when the patent intercostal or lumbar artery that fed the AA was covered [25]. Preoperative identification of the feeding artery of the AA by selective spinal arteriography has been proposed by other groups [10,26,27,28,29]. Briefly, their procedure consisted of selective catheterization, usually by the femoral route, followed by the manual injection of contrast material for the imaging of intercostal and lumbar arteries until the arteries that supplied the anterior spinal artery were identified.

However, treatment of a DTA aneurysm might imply a voluntary occlusion of the feeding artery of the AA to avoid the risk of aortic rupture. Several methods have been developed to decrease the risk of spinal cord injury during treatment of a DTA aneurysm. Sequential treatment avoiding extensive covering of the intercostal artery and collateral of interest, such as the left subclavian artery or hypogastric artery, is recognized as an efficient method [30]. Preoperative coiling of the lumbar arteries is also described as a solution to reduce complications [31,32]. Curative treatment of spinal cord ischemia has been shown to be effective [4,29,33,34]. The goal of the treatment is to increase the medullary perfusion pressure and to increase the development of the collateral circulation. Banga et al. [35] increased the mean arterial blood pressure (>80–90 mmHg) and used cerebrospinal fluid drainage to increase the medullary perfusion pressure. Assessment of motor evoked potentials allows the monitoring of the blood supply to the spinal cord in real time during intervention and the adaptation of the arterial blood pressure to cerebrospinal fluid drainage. This setting has been shown to reduce postoperative complications [5,36]. In our clinical practice, all these methods are used and personally adapted according to the patient data. We thought that the precise location of the feeding artery of the AA could help us to better decide on neurological risk reduction techniques during surgery.

Recent innovations in CT or MR technology have made it possible to noninvasively identify the feeding artery of the AA. The detection rates for this artery have been reported to be 80–90% using CT or MR angiography [25,26,27,28,29,33] in patients with thoracic aneurysm or dissection. In fact, these noninvasive assessment methods are able to depict the morphologic hairpin turn configuration of the AA. However, the entire course of the AA should be identified by demonstrating continuity from the anterior spinal artery to the aorta via the AA to avoid misinterpretation of the arterial vascularization of the spinal cord. The rate for vascular continuity has been reported to be in the range of 25 to 60% for 16- or 64-detector row CT [12,14,15,17]. The difficulty in demonstrating continuity can be attributed to the small size of the artery and the proximity to the spine [37]. It is also important to emphasize that many studies have been conducted in different ethnic groups [10,23,28], whose physical characteristics differ from our population.

This study has some limitations, especially the small number of patients included. Angiography is invasive and requires radiation exposure, iodine chelate injection into the patient, and medical expertise and time. This was a proof-of-concept study, and we did not prove that the proposed method is able to decrease the occurrence of spinal cord injury.

## 5. Conclusions

These results suggest that the combined use of angio-CBCT and 3D road-mapping to guide the anatomical location of the feeding artery of the AA was feasible in all the patients included in our series. In our team’s experience, precise knowledge of the arterial vascularization of the spinal cord allowed us to modify and personalize the treatment of DTA diseases.

## Figures and Tables

**Figure 1 jpm-12-01890-f001:**
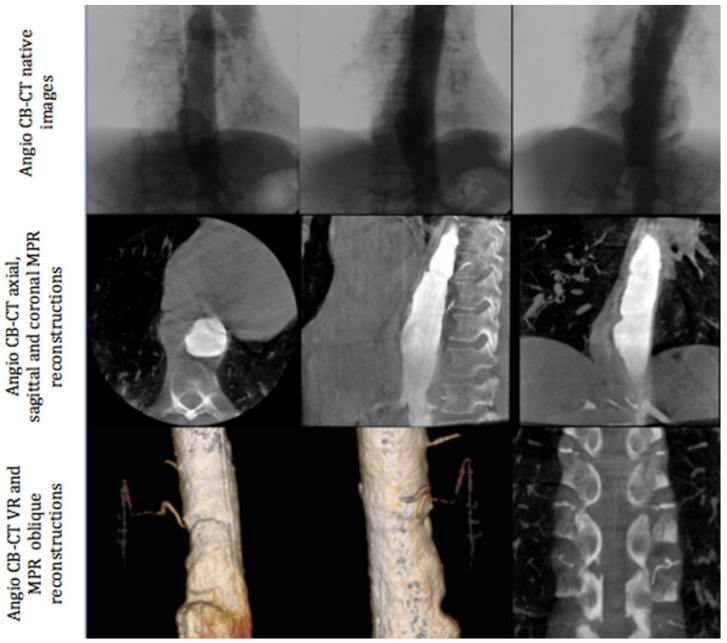
Different phases of angio-CBCT image acquisition: native images (upper section), MPR reconstructions (middle section), and volume rendering and oblique reconstructions (lower section).

**Figure 2 jpm-12-01890-f002:**
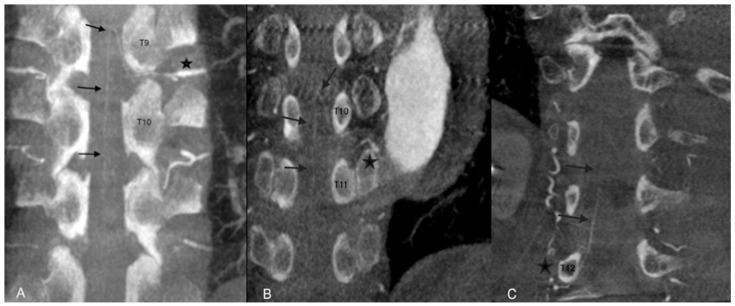
Examples of angio-CBCT reconstructions showing the Adamkiewicz artery (AA) (black arrows) in three different patients. The AA is seen running obliquely along the anterior surface of the spinal cord with a classic hairpin turn and connection to the anterior spinal artery (star). In cases (**A**,**B**), the feeding artery of the AA was clearly depicted. In case (**C**), complementary selective arteriography was performed.

**Figure 3 jpm-12-01890-f003:**
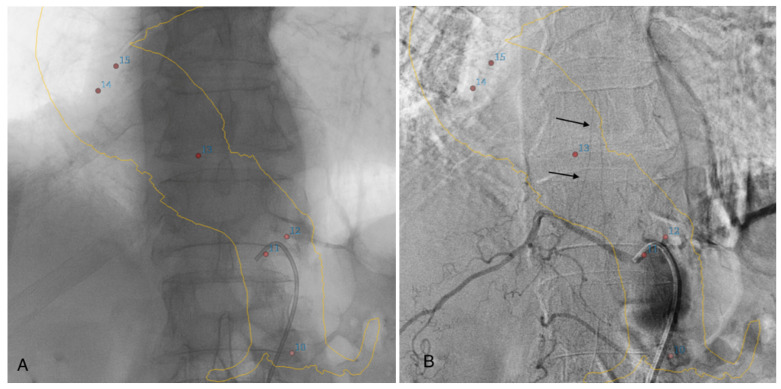
Three-dimensional road-mapping of the intercostal and lumbar artery ostia (red dots) and aortic wall mask (yellow lines), extracted from angio-CBCT acquisition (**A**). Selective angiography of a patent intercostal artery with visualization of the classic hairpin turns (black arrows) (**B**).

**Figure 4 jpm-12-01890-f004:**
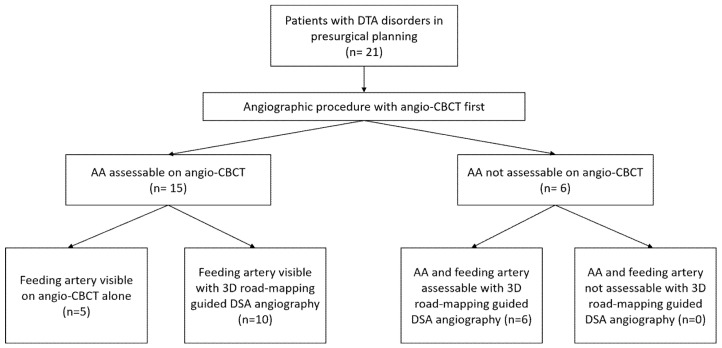
Study flow chart. DTA: descending thoracic aorta; AA: Adamkiewicz artery.

**Table 1 jpm-12-01890-t001:** Population characteristics (*n* = 21).

Variable	Patient Characteristics
Age, years (mean ± SD)	68 ± 11
Male, *n* (%)	17 (81)
BMI, kg/m² (mean ± SD)	27 ± 3.5
Hypertension, *n* (%)	17 (81)
Hyperlipidemia, *n* (%)	7 (33)
Diabetes, *n* (%)	2 (9)
Smoking, *n* (%)	14 (66)
Descending thoracic aorta pathology, *n* (%)	
Aneurysm	19 (90)
Dissection	2 (10)
Aortic diameter, mm (mean ± SD)	61.6 ± 8.1

**Table 2 jpm-12-01890-t002:** Results of angiographies according to the visualization of the anterior spinal artery on the CBCT acquisition (*n* = 21). AA: Adamkiewicz artery.

	AA Assessable on CBCT (*n* = 15)	No AA Assessable on CBCT (*n* = 6)	*p*
Second angio-CBCT, *n* (%)	2 (13%)	6 (100%)	
Feeding artery visible on angio-CBCT, *n* (%)	5 (33)	x	
Feeding artery visible after selective guided DSA catheterization, *n* (%)	10 (67)	6 (100)	
Amount of iodine, mL (mean ± SD)	71.8 ± 38.7	90.0 ± 26.1	0.23
Fluoroscopy time, min (mean ± SD)	16.7 ± 10.8	13.1 ± 8.7	0.45
Length of the procedure, min (mean ± SD)	46.5 ± 17.2	47.9 ± 26.0	0.84
Irradiation, Gy/cm^2^ (med (IQR))	25.5 (20.0–40.0)	70.4 (30.8–96.9)	0.41
Aortic aneurism diameter, mm (mean ± SD)	60.33 ± 7.6	64.6 ± 9.3	0.28

**Table 3 jpm-12-01890-t003:** Distribution of the side and the level of the origin of the Adamkiewicz artery (AA) (*n* = 21).

	Right (*n* = 6)	Left (*n* = 15)
T7, *n* (%)	0 (0)	1 (5)
T9, *n* (%)	2 (10)	4 (20)
T10, *n* (%)	1 (5)	2 (10)
T11, *n* (%)	1 (5)	6 (25)
T12, *n* (%)	1 (5)	1 (5)
L1, *n* (%)	0 (0)	1 (5)
L3, *n* (%)	1 (5)	0 (0)

**Table 4 jpm-12-01890-t004:** Procedure modifications following presurgical identification of the AA for both endovascular and open surgery treatments and neurological complication follow-up (*n* = 16). CSF: cerebrospinal fluid; AA: Adamkiewicz artery.

	Endovascular (*n* = 11)	Open Surgery (*n* = 5)
Modification of the surgery, *n* (%)	8 (73%)	3 (60%)
- Stent graft length reduction, *n* (%)	5 (45%)	x
- Monitoring/CSF drainage, *n* (%)	7 (64%)	2 (40%)
- AA reimplantation, *n* (%)	x	3 (60%)
Neurologic complication, *n* (%)	0	1 (20%)

## Data Availability

Not applicable.

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
