# Peer review of "Angio Cone-Beam CT (Angio-CBCT) and 3D Road-Mapping for the Detection of Spinal Cord Vascularization in Patients Requiring Treatment for a Thoracic Aortic Lesion: A Feasibility Study"

_jpm, 2022, doi:10.3390/jpm12111890_

Round 1

Reviewer 1 Report

This study employed angio CBCT to identify the feeding artery of the Adamkiewicz artery in patients with descending thoracic aorta aneurysms or dissections. The study objectives are well defined at the end of Introduction, but the contents don’t fulfill the goals, there are some concerns not appropriately addressed by the current manuscript.

Introduction:

1.     The technical uniqueness of angio CBCT was not fully explained in the Introduction part, making the general readers not fully understand the novelty of the methodology.

2.     The significance of locating AA in the clinical practice was not fully illustrated.

Methods:

1.     Pre-operative angio CT and XA were fused using anatomical bone landmarks manually: was this method reliable? The tiny structure of small vessels could have relative displacement to the bones along with the breathing movement.

2.     A major limitation of the study is the small sample size of 21 cases. The patients were from 02/2018 to 04/2020 in the Centre. Why not include cases from the last two years?

3.     In case of a lack of opacification of the feeding artery of the AA, a second angio CBCT was performed at a different level of the aorta. So what was the occurrence rate of a second angio CBCT? Would this increase the radiation dose significantly? This should be included in the analysis and discussion.

Results:

1.     Line 208 reported the actual modification of the following procedure. This is key information regarding the utility of the detection of AA by CBCT, yet the exact rationale of the modification was not illustrated. How the different results of CBCT would change the following procedure should be described in advance in the introduction and methods.

2.     It’s mentioned in the objectives and methods that patients had 1 month follow-up evaluation. Yet the results of the 1 month follow-up were not systematically reported. It was mentioned that one patient experienced a post procedural spinal cord injury, but when did it happen (I month later?) and whether it is the only case of poor prognosis were not reported.

Discussion:

1.     Some statements were not supported by the results of the current study, which was small sampled without a control group. Such as “preprocedural location of the patent ostium of intercostal or lumbar artery avoids unnecessary catheterization attempts”, or “it might play a role in planning of surgical strategy to decrease the risk of neurological complication” (the complication was not systematically reported in the experimental group or the control group).

2.     In fact, many studies have performed preoperative identification of the feeding artery of the AA by selective spinal arteriography [8,20–23], and detection rates for this artery have been reported to be 80 - 90 % using CT or MR angiography [19–23,27] in patients with thoracic aneurysm or dissection. The incremental value of angio CBCT might be the potential to identify the vascular continuity at a higher rate. Yet it was not discussed how the vascular continuity improved the following clinical management compared to detecting the artery using CT or MR angiography alone. 

Table 2:

Why the SD of the irradiation so large? 118.2 ± 135.4 in the No AA on CBCT group.

Author Response

REVIEWER 1

This study employed angio CBCT to identify the feeding artery of the Adamkiewicz artery in patients with descending thoracic aorta aneurysms or dissections. The study objectives are well defined at the end of Introduction, but the contents don’t fulfill the goals, there are some concerns not appropriately addressed by the current manuscript.

Response: thank  you for your comments and review. Following your advices, global modifications have been made in the manuscript to improve the manuscript readability.

First, the term of 2D/3D fusion has been removed from the manuscript. We conduct an angio-CBCT for all patients, to visualize of AA. We also used angio CBCT as a 3D road mapping to localize the ostium of all patent intercostals or lumbar artery before selective DSA. Then, we described our results in terms of AA opacification, irradiation and iodine contrast dose. And how the localization of AA impacts our surgical planning.

This study is a feasibility study with descriptive results, we have added these terms in the title, objectives and M&M section. We wanted to describe the feasibility of this technique, its results in terms of AA visualization and procedural technical parameters.

Introduction:

  1. The technical uniqueness of angio CBCT was not fully explained in the Introduction part, making the general readers not fully understand the novelty of the methodology.

Response:  following your comment, precisions have been added in the introduction section. We have added the following paragraph in introduction section : “The peri-procedural use of cone beam CT (CBCT) has been described in many interventional radiology procedures [18]. The arm of the angio-suite can perform localized CT acquisition by rotating around the patient in a cone-shaped beam. The acquisition, coupled with vessel angiography is called angio-CBCT. It allows small vessels detection with a higher spatial resolution than CT–angiography and MRI acquisition. Also, angio CBCT acquisition can be used as a 3D road mapping mask for the rest of the procedure. It was found efficient in the detection of injured vessels in emergency embolization, transarterial chemoembolization guidance or evaluation of spinal arteriovenous fistula [19] [20] [21]. There is no evaluation of this technique in the pre-procedural visualization of the AA feeding arteries before DTA disease treatment. »       

  1. The significance of locating AA in the clinical practice was not fully illustrated.

Response: following your comment, modifications have been made in the introduction section to precise the need and significance for AA localization in clinical practice.  The following paragraph have been added : “The need for assessing anatomical location of the feeding artery of the (AA) before DTA treatment is debated in the literature for several reasons. The  variability of its anatomical origin is high, most commonly found between the T8 and L1, and originates from left intercostal or lumbar artery in 70% to 85% of cases [6,7]. Also, the feeding artery of AA is a small artery and its anatomical location might be hidden with the deformation of the aneurysmal aorta and aortic thrombus. However, a lower rate of post operative neurologic complications is observed in patients with DTA surgical repair if they beneficiate from a previous AA detection and preservation [8]. If AA feeding artery is covered in TEVAR, CSF drainage is associated with lower symptomatic spinal cord ischemia incidence [9]. There is no consensus about spinal cord ischemia prevention before descending thoracic aorta surgical treatment. Anatomical location of AA might help in decision making before treatment. Selective DSA of each patent intercostals artery could be very challenging in case of large aortic aneurysm. New angiographic techniques such as angio CBCT might help in detecting small arteries such as AA in large aneurismal vessel »

Methods:

  1. Pre-operative angio CT and XA were fused using anatomical bone landmarks manually: was this method reliable? The tiny structure of small vessels could have relative displacement to the bones along with the breathing movement.

Response: we agree with the reviewer that the term of 2D/3D fusion is confusing and have been removed from the revised version. We changed in the whole manuscript the term 2D/3D fusion by angio-CBCT 3D road-mapping. We actually mean that when AA vascular continuity was not assessable with Angio CBCT alone, we performed a 3D VR reconstructions of the angio CBCT acquisition and localized lumbar and thoracic arteries ostia using a 3D road –mapping. In these cases, we have had a perfect correlation between angio-CBCT road map and anatomy.

  1. A major limitation of the study is the small sample size of 21 cases. The patients were from 02/2018 to 04/2020 in the Centre. Why not include cases from the last two years?

Response: since April 2020, our recruitment was severely impacted by COVID-19 crisis due to hospital organization. Since we began to write this manuscript, we did not gather all dedicated information for the patients we treated. We could not include more patient in the present study.

  1. In case of a lack of opacification of the feeding artery of the AA, a second angio CBCT was performed at a different level of the aorta. So what was the occurrence rate of a second angio CBCT? Would this increase the radiation dose significantly? This should be included in the analysis and discussion.

Response: a second angio-CBCT was performed in 2/15 cases in the group “AA assessable on CBCT” when the first angio-CBCT was not sufficient ot assess AA anatomy and vascular communication. A second angio CBCT was performed in 6/6 of the “no assessable AA on CBCT. Informations have been added in the Table 2. 

  1. Line 208 reported the actual modification of the following procedure.This is key information regarding the utility of the detection of AA by CBCT, yet the exact rationale of the modification was not illustrated. How the different results of CBCT would change the following procedure should be described in advance in the introduction and methods.

Response:  the direct impact of angio-CBCT acquisition in surgical planning is not under the scope of that descriptive study

Our objective (objective 3 at the end of the introduction) was to describe how the detection of the Adamkiewicz artery modified our surgical strategy. Our surgical strategy was based on the most complete preservation of the spinal vasculature, given the absence of international recommendations or consensus. Modifications were made in the abstract and the material and methods section to clarify this objective.

It was therefore decided that if the AA was born in a high-risk area, we would modify the surgery to preserve the vascularization as best as possible. We have described  in MM section our attitude towards surgical modifications depending on AA localization (M&M section - 2.3 “surgical management”). These results are shown in Table 4. 

  1. It’s mentioned in the objectives and methods that patients had 1 month follow-up evaluation. Yet the results of the 1 month follow-up were not systematically reported. It was mentioned that one patient experienced a post procedural spinal cord injury, but when did it happen (I month later?) and whether it is the only case of poor prognosis were not reported.

Response:  one-month follow-up was only concerning post-procedural neurological complication (spinal cord ischemia, paresthesia or motor disorders of the lower limb or ponytail syndrome). Precisions have been made in the Material and methods section.

We recorded all-causes of post-porcedural complications but did not find necessary to present them, since we want to present surgery planification throught the theme of neurologic strategy planning. We had one neurologic complication (spinal cord ischemia) in the open surgery group, as presented in the Table 4. This complication occurred immediately in the close post-procedure follow-up in the recovery room, and was still present at one month, confirmed by spinal MRI.

Discussion:

  1. Some statements were not supported by the results of the current study, which was small sampled without a control group. Such as “preprocedural location of the patent ostium of intercostal or lumbar artery avoids unnecessary catheterization attempts”, or “it might play a role in planning of surgical strategy to decrease the risk of neurological complication” (the complication was not systematically reported in the experimental group or the control group).

Response:  thank you for these comments, following your advise we have removed these confusing sentence from the revised manuscript.

  1. In fact, many studies have performed preoperative identification of the feeding artery of the AA by selective spinal arteriography [8,20–23], and detection rates for this artery have been reported to be 80 - 90 % using CT or MR angiography [19–23,27] in patients with thoracic aneurysm or dissection. The incremental value of angio CBCT might be the potential to identify the vascular continuity at a higher rate.

Response:  thank you for your comment, The literature about MR and CT for AA detection arize from asia. In our country ethnicity and risk factor of thoracic aneurysm modified patient morphology. All non invasive method to assess AA are not applicable with the same results in our population explaining why we work on CBCT.

  1. Yet it was not discussed how the vascular continuity improved the following clinical management compared to detecting the artery using CT or MR angiography alone. 

Response: we fully agree with that comment that part was not under the scope of the present paper. We could not address that point and that comment have been added in the limitation section

Table 2:

Why the SD of the irradiation so large? 118.2 ± 135.4 in the No AA on CBCT group.

Response: In this group, two of the patients gathered all pejorative situations leading to abnormally  high irradiation: 2 angio-CBCT were initially necessary as described in the Material and Methods section. Patients had  high BMI (36 kg/m² and 41 kg/m²)) which lead to an increase in fluoroscopic and angiographic acquisition. Also, arterial catheterization was long and difficult due to large aneurysms and a complex anatomy. This lead to a significantly high irradiation dose in these particular patients explaining the large standard deviation in this group.

Reviewer 2 Report

The authors reported their surgical patient series of descending thoracic artery aneurysms using cone beam CT (CBCT) and elective DSA guided by image fusion (IF) to detect the Adamkiewicz artery. IF is a technology that improves anatomical understanding during endovascular procedures without the need to perform digital subtraction angiography. CBCT provides a three-dimensional rendering of opacified vascular structures and, accompanied by IF, serves as a useful guide for endovascular aortic procedure. Since the detection of AA is an essential point in the prevention of spinal cord ischemia, which is the biggest complication after TAA surgery, it will be interesting to see if this method can improve the detection rate. This study was performed retrospectively, is a single arm of a small number of cases, and is rather descriptive.

Major comment

The authors used this strategy to achieve preoperative Adamkiewicz artery identification in all 21 cases in their series. The authors showed good results, with only one complication in the 16 cases that were operated on. In my opinion, this is a report well worth reporting. However, a more detailed comparison of the identification rate with the conventional method should be provided.

Minor comment

In table 2, the percentage of AA assessable on selective DSA should be 67.

In line 228, what complication are you referring to, spinal cord ischemia or not?

Author Response

REVIEWER 2

The authors reported their surgical patient series of descending thoracic artery aneurysms using cone beam CT (CBCT) and elective DSA guided by image fusion (IF) to detect the Adamkiewicz artery. IF is a technology that improves anatomical understanding during endovascular procedures without the need to perform digital subtraction angiography. CBCT provides a three-dimensional rendering of opacified vascular structures and, accompanied by IF, serves as a useful guide for endovascular aortic procedure. Since the detection of AA is an essential point in the prevention of spinal cord ischemia, which is the biggest complication after TAA surgery, it will be interesting to see if this method can improve the detection rate. This study was performed retrospectively, is a single arm of a small number of cases, and is rather descriptive.

Response: Thank you for our comments and review We fully agree with you this is a feasibility study and this have been added in the revised version of the manuscript.

Major comment

The authors used this strategy to achieve preoperative Adamkiewicz artery identification in all 21 cases in their series. The authors showed good results, with only one complication in the 16 cases that were operated on. In my opinion, this is a report well worth reporting. However, a more detailed comparison of the identification rate with the conventional method should be provided.

Response : following your advise we have added a paragraph in Discussion section explainging the results obtained in previous study in the literature using DSA.  

Minor comment

In table 2, the percentage of AA assessable on selective DSA should be 67.

Response: following to your comments, modifications have been made in the final version of the manuscript.

In line 228, what complication are you referring to, spinal cord ischemia or not?

Response: Yes, precision has been made in the manuscript.

Round 2

Reviewer 1 Report

The manuscript has been modified globally and improved with more clarity and scientific soundness, and some unaddressed issues have been answered. I have three remaining questions:

1.     The last paragraph described the 1-month follow-up results of included patients. It should be clarified if the reported one case of poor prognosis was the only one.

2.     The irradiation dose in Table 2: has the normality test be performed? The mean-SD was less than zero in the “No AA assessable on CBCT” group, probably because the data was skewed. In such circumstance, median (IQR) should be reported instead of mean±SD.

3. It is still confusing to see in the responses that "all non invasive method to assess AA are not applicable with the same results in our population". 

Author Response

REVIEWER 1 :

The manuscript has been modified globally and improved with more clarity and scientific soundness, and some unaddressed issues have been answered. I have three remaining questions:

  1. The last paragraph described the 1-month follow-up results of included patients. It should be clarified if the reported one case of poor prognosis was the only one.

RESPONSE: Thank you for your comments. This was the only neurological complication. As described in the M&M section, the one month follow-up was only dedicated (for this study) to neurological complications. There was only one, which is described in the results section. For information, we encountered a significant pleural effusion in the open surgery group that needed a drainage. In the endovascular groups, we had two groin hematoma, one that was resolved with external compression alonen one that needed a covered stent placement at  H24. These complications were

  1. The irradiation dose in Table 2: has the normality test be performed? The mean-SD was less than zero in the “No AA assessable on CBCT” group, probably because the data was skewed. In such circumstance, median (IQR) should be reported instead of mean±SD.

RESPONSE: Following your advices we changed the data in Table 2 and presented the irradiation dose in terms of median (IQR) values instead of mean ±SD..

  1. It is still confusing to see in the responses that "all non invasive method to assess AA are not applicable with the same results in our population". 

RESPONSE: Thank you for your comments.It is true that this sentence is not clear enough. By this we meant to explain that previous literature in this research field was mainly in Asian population for the most cited papers (1) (2)  (3) (4) (5) (6) (7) (8). This was to justify our study: in a way, our study is not the  first one to evaluate this, but, an evaluation in our population is necessary. Also we cannot transpose non invasive methods results to our population, since we can admit our population is not identicial to the one on which the previous researches were focused.

The term “all non invasive” is certainly false and should be replace by “most non invasive methods that have been evaluate to assess AA were not tested in our population. Given the differences in terms of population characteristics, we cannot transfer these results to our population.”

  1. Kudo K, Terae S, Asano T, et al. Anterior spinal artery and artery of Adamkiewicz detected by using multi-detector row CT. AJNR Am J Neuroradiol. 2003;24(1):13–17.
  2. Hino T, Kamitani T, Sagiyama K, et al. Detectability of the artery of Adamkiewicz on computed tomography angiography of the aorta by using ultra-high-resolution computed tomography. Jpn J Radiol. 2020;38(7):658–665. doi: 10.1007/s11604-020-00943-3.
  3. Kodama Y, Sakurai Y, Yamasaki K, Yamada G. Detection of arteriography-negative anterior spinal artery branching via intercostobronchial trunk confirmed by CT during intercostobronchial trunk arteriography: A case report. Radiol Case Rep. 2020;15(7):832–836. doi: 10.1016/j.radcr.2020.03.024.
  4. Nishii T, Kotoku A, Hori Y, et al. Filtered back projection revisited in low-kilovolt computed tomography angiography: sharp filter kernel enhances visualization of the artery of Adamkiewicz. Neuroradiology. 2019;61(3):305–311. doi: 10.1007/s00234-018-2136-8.
  5. Yoshioka K, Niinuma H, Ehara S, Nakajima T, Nakamura M, Kawazoe K. MR Angiography and CT Angiography of the Artery of Adamkiewicz: State of the Art. RadioGraphics. 2006;26(suppl_1):S63–S73. doi: 10.1148/rg.26si065506.
  6. Chatani S, Haimoto S, Sato Y, et al. Preoperative Embolization of Spinal Metastatic Tumor: The Use of Selective Computed Tomography Angiography for the Detection of Radiculomedullary Arteries. Spine Surg Relat Res. 2021;5(4):284–291. doi: 10.22603/ssrr.2020-0202.
  7. Uotani K, Yamada N, Kono AK, et al. Preoperative Visualization of the Artery of Adamkiewicz by Intra-Arterial CT Angiography. Am J Neuroradiol. 2008;29(2):314–318. doi: 10.3174/ajnr.A0812.
  8. Hyodoh H, Kawaharada N, Akiba H, et al. Usefulness of Preoperative Detection of Artery of Adamkiewicz with Dynamic Contrast-enhanced MR Angiography. Radiology. 2005;236(3):1004–1009. doi: 10.1148/radiol.2363040911.